# An analysis of knowledge, attitude and practice gaps in scientific buffalo husbandry

Gururaj Makarabbi◉, Aiswarya Sabu◉*, Navneet Saxena, Madan Lal Sharma

Indian Council of Agricultural Research–Central Institute for Research on Buffaloes (ICAR–CIRB), Hisar, Haryana, India

◉ These authors contributed equally to this work.
* aishuambady@gmail.com

## Abstract

Buffalo husbandry remains vital to India's dairy sector, yet adoption gaps hinder productivity. This study assesses knowledge, attitudes, and practices (KAP) among 380 buffalo farmers in India, revealing critical disparities. While awareness was relatively high in key areas—heat detection (79.4%), mastitis identification (77.4%), and balanced feed (77.9%)—practice rates lagged severely (32.9–44.7%), with the widest gap in silage-making (71.5% awareness vs. 32.9% adoption, a 38.6-point deficit). Cluster analysis identified five farmer segments, highlighting adoption disparities: smallholders (36.8% of the sample) practised only 25% of techniques despite moderate knowledge, while commercial farmers (7.9%) achieved 90.5% adoption. Chi-square tests confirmed education (OR=3.4, p<0.001) and herd size (OR=2.4, p<0.01) as the strongest KAP predictors, with logistic regression under-scoring income's pivotal role (OR=4.3 for practices). In order to bridge gaps, the study proposes three strategies: (1) targeted training for smallholders through farmer groups, (2) subsidized input bundles (feed, minerals) paired with mobile-based advi-sories, and (3) market incentives for quality milk to reward adoption. Findings stress the need to replace blanket extension approaches with segmented interventions addressing structural barriers (credit, vet access) and attitudinal resistance (silage scepticism, vaccine hesitancy). Policy action should prioritize doubling frontline extension staff and integrating cooperatives with digital tools to scale solutions.

## Introduction

Dairy farming occupies a central position in India's agricultural economy, serving as a crucial source of livelihood for millions of rural households while making significant contributions to the nation's food and nutritional security. India is ranked 1st in milk production since 1998, now contributing 25 percent of global milk production [1], with buffaloes playing an outsized role by contributing approximately 48% of the country's total milk output despite constituting only 21% of the milch animal population [2].

**Data availability statement:** The dataset can be found at the following DOI: https://doi.org/10.3886/E234181V1.

**Funding:** The author(s) received no specific funding for this work.

**Competing interests:** The authors have declared that no competing interests exist.

The sector's socioeconomic importance cannot be overstated – it provides year-round income to small and marginal farmers, employs a significant proportion of rural women, and serves as a vital hedge against crop failures [3,4]. However, this critical sector faces persistent productivity challenges, with average buffalo milk yields stagnating at 4–5 litres per day compared to the 8–10 litres achievable through improved management practices [5].

The disconnect between available scientific knowledge and actual farming practices – known as the Knowledge-Attitude-Practice (KAP) gap – represents a significant constraint in realizing the full potential of buffalo husbandry. While research institutions have developed numerous improved practices ranging from balanced feeding regimens to advanced reproductive technologies, their adoption remains inconsistent at the field level [6–8]. This implementation gap is particularly pronounced in buffalo systems, which have historically received less research and extension attention than cattle husbandry [9,10]. Existing studies highlight several dimensions of this challenge: farmers often possess theoretical knowledge about scientific practices but fail to implement them due to financial constraints, lack of access to inputs, or deeply ingrained traditional methods [11–13]. More fundamentally, there remains a limited understanding of how farmers' attitudes and perceptions mediate between knowledge acquisition and practice adoption – a critical gap this study addresses.

Understanding KAP dynamics in buffalo husbandry is essential for multiple reasons. First, with the growing demand for milk and milk products, bridging yield gaps through improved practices could enhance productivity by 30–40% without expanding herd sizes [14]. Second, systematic KAP analysis helps identify the most significant barriers to technology adoption, enabling more targeted and effective extension strategies [15–17]. Third, as climate change pressures intensify, promoting resilient buffalo husbandry practices becomes crucial for sustaining rural livelihoods [18–20]. Despite this importance, current research suffers from three key limitations: most studies focus on single practices rather than holistic KAP assessment [21–23] few examine the attitudinal dimensions that influence adoption decisions [24–26] and minimal work has been done to segment farmers based on their adoption potential and constraints [27–29].

Our study addresses these research gaps by comprehensively examining KAP patterns in scientific buffalo husbandry. The research builds on existing literature while introducing several novel elements: it adopts an integrated framework that simultaneously assesses knowledge levels, attitudinal predispositions, and practice adoption across multiple husbandry domains; incorporates advanced cluster analysis to identify distinct farmer segments; and examines how socioeconomic factors interact with knowledge and attitudes to shape adoption behaviours. The study is guided by four key research questions: (1) What is the magnitude of knowledge-practice gaps across different buffalo husbandry domains? (2) How do farmers' attitudes mediate between knowledge acquisition and practice adoption? (3) What distinct farmer segments emerge based on their KAP profiles and characteristic constraints? (4) Which socioeconomic factors most strongly influence practice adoption patterns? By answering these questions, the study provides theoretical insights into technology

adoption processes and practical guidance for designing more effective extension interventions tailored to different farmer categories.

The findings are relevant to ongoing policy initiatives like the Rashtriya Gokul and the National Livestock Mission, offering evidence-based strategies to improve extension delivery and technology uptake. From a theoretical perspective, the study contributes to the broader literature on agricultural innovation systems by demonstrating how KAP analysis can reveal the complex interplay between knowledge, perceptions, and implementation barriers in livestock production systems. At a practical level, the farmer segmentation approach provides a model for developing targeted extension strategies that recognize the heterogeneity of smallholder production systems – a crucial requirement for achieving sustainable intensification of buffalo husbandry in India.

## Methodology

### Sample and procedure

The study adopted a cross-sectional design, gathering data from 380 buffalo farmers across India using purposive sampling. The selection criteria included (1) ownership of at least one milking buffalo, (2) a minimum of one year of buffalo-rearing experience, and (3) dairy farming as a primary livelihood.

The interview schedule was divided into two key sections. The first section collected socio-demographic and farm characteristics, including personal details (name of the respondent, age, gender, education level, family size), economic indicators (primary occupation, monthly income), and production metrics (herd size). The second section assessed knowledge, attitude and adoption of identified scientific buffalo husbandry practices

Data collection was conducted via face-to-face interviews by trained enumerators proficient in local dialects, with each session lasting 45–60 minutes. The interview schedule was pretested with 40 farmers to refine clarity and cultural appropriateness, achieving strong reliability.

### Ethical considerations

The study was reviewed by the Institute Animal Ethics Committee (IAEC) of ICAR–Central Institute for Research on Buffaloes (ICAR–CIRB), Hisar. Based on the nature of the research—non-invasive, non-clinical, and involving only adult participants engaged in agricultural activities—the committee determined that formal ethical approval was not required, and thus the study was exempted from full review. The research involved face-to-face interviews with buffalo farmers, collecting information on socio-demographic characteristics and the adoption of scientific buffalo husbandry practices. The study posed no foreseeable physical, psychological, or social risks to participants. Prior to each interview, verbal informed consent was obtained after explaining the purpose of the study, assuring confidentiality, and informing participants of their voluntary involvement. No personal identifiers were included in the dataset, and respondent anonymity was maintained throughout data handling and analysis. At the end of each interview, a signature or thumb impression was obtained from the participant directly on the interview schedule as a confirmation of their voluntary participation.

### Selection criteria and reliability assessment of key buffalo husbandry practices for KAP analysis

The study assessed eight critical scientific buffalo husbandry practices that were carefully selected based on their documented impact on productivity, animal health, and farm profitability (Table 1). These practices represent fundamental pillars of improved buffalo management that have been consistently emphasized in livestock extension programs but show varying adoption rates across smallholder systems. Nutrition-related practices (balanced feed preparation, mineral mixture formulation, and silage making) were included due to their proven ability to enhance milk yield by 25–40% while reducing feed costs [30]. Despite this evidence, previous studies indicate only 30–45% adoption among smallholders, primarily due to knowledge gaps and input accessibility issues [30,31]. Health management practices (sick animal identification and

**Table 1. Scientific buffalo husbandry practices selected for assessment.**

| Practice Category | Specific Practices | Rationale for Inclusion | Key References |
|---|---|---|---|
| Nutritional Management | Balanced feed preparation | Improves feed efficiency by 30% and milk fat content | [30] |
| | Mineral mixture formulation | Addresses widespread micronutrient deficiencies affecting reproduction | [38] |
| | Silage making | Reduces dry season feed shortages by 40% | [39] |
| Health Management | Sick animal identification | Enables early disease intervention, reducing mortality by 25% | [40] |
| | Mastitis detection | Prevents 15–20% production losses in affected animals | [32] |
| Reproductive Management | Heat symptom identification | Critical for optimal breeding timing (21-day cycle) | [34] |
| Genetic Improvement | High-yielder identification | Enables 18–22% higher lifetime productivity through selective breeding | [41] |
| | Murrah buffalo recognition | Purebred animals yield 3.5-4.5L more milk/day than non-descript varieties | [36] |

mastitis detection) were prioritized, given that mastitis causes 26% of production losses in smallholder herds [32]. Their inclusion follows [33], recommendations that early disease detection can reduce treatment costs by 60% while improving animal welfare. Reproductive management (heat symptom identification) was selected as delayed breeding costs farmers ₹150–200/day in lost productivity [34], with studies showing that only 35% of farmers accurately detect estrus [35]. Genetic improvement practices (identifying high-yielders and Murrah buffaloes) were incorporated because breed selection contributes 40% of genetic gain potential [36]. However, field data suggests <50% of farmers apply selection criteria [37]. Pilot testing (n = 30) confirmed the instrument's reliability, achieving Cronbach's α = 0.84 for knowledge items and α = 0.81 for attitude scales, indicating high internal consistency.

## Measurement of knowledge, attitudes, and practices

The study employed a comprehensive methodology to assess farmers' knowledge, attitudes, and practices (KAP) regarding scientific buffalo husbandry. Knowledge was measured through a binary response system where farmers were asked to indicate their awareness of eight key practices, scoring 1 for "aware" (correct description when probed) and 0 for "unaware" (no knowledge or incorrect information). This yielded a total knowledge score ranging from 0–8, with higher scores indicating greater awareness. Practice adoption was evaluated through self-reported data, where farmers received a score of 1 for correctly implementing practice and 0 for non-adoption, resulting in a similar 0–8 practice score range. Attitude assessment utilized a 5-point Likert scale (0 = strongly disagree to agree 4 = strongly) across twelve statements covering traditional versus scientific methods, nutrition and health management, and economic considerations. The attitude scale demonstrated strong internal consistency (Cronbach's α = 0.82), with positive statements directly scored and negative statements reverse-coded to ensure uniform interpretation.

## Data cleaning

All 398 interview schedules were fully completed (no missing values). Using Mahalanobis distance (p < 0.001), 18 multivariate outliers (2.3%) were identified and excluded, yielding a final sample of 380.

## Data analysis

The collected data were systematically organized in Microsoft Excel and analyzed using descriptive and inferential statistical methods. Knowledge scores for each respondent were calculated as the percentage of correct responses out of the total knowledge questions [(Number of correct answers/ Total questions) × 100]. Similarly, practice adoption scores were computed as the percentage of scientifically recommended practices actually implemented by farmers [(Number of practices adopted/ Total practices assessed) × 100]. For attitude assessment, the mean score across all Likert-scale statements was derived to evaluate farmers' overall perceptions toward improved buffalo husbandry techniques. To visualize

disparities between awareness and implementation, quadrant analysis was performed, categorizing practices into four groups: (1) high knowledge-high adoption, (2) high knowledge-low adoption, (3) low knowledge-high adoption, and (4) low knowledge-low adoption. Hierarchical cluster analysis (Ward's method with Euclidean distance) was then applied to classify farmers into groups based on their KAP profiles, revealing adoption patterns across different typologies (e.g., resource-constrained smallholders vs. commercial farmers). For inferential analysis, binary logistic regression identified key predictors (e.g., education, herd size, income) of three binary outcomes: (1) knowledgeable vs. non-knowledgeable status, (2) good vs. poor practices, and (3) positive vs. negative attitudes. Additionally, chi-square tests examined associations between demographic variables (age, gender, experience) and KAP components, with odds ratios (OR) and 95% confidence intervals (CI) quantifying effect sizes for significant relationships ($p < 0.05$). All analyses were conducted in R (v4.4.3) and SPSS (v26), ensuring robust validation of results.

## Result

The study assessed farmers' awareness (Fig 1) of key scientific buffalo husbandry practices across several domains. In reproductive management, 79.41% of respondents demonstrated knowledge of heat symptom identification. For animal health management, 75.59% showed awareness of sick animal identification, while 77.35% were familiar with mastitis detection methods. Nutrition-related knowledge showed slightly different patterns. Awareness of balanced feed preparation methods was reported by 77.94% of farmers, while 71.76% knew about mineral mixture preparation. Knowledge of silage-making techniques was evident among 71.47% of respondents. Regarding breed identification and selection, 77.94% of participants could correctly identify Murrah buffaloes, and 74.12% reported recognising high milk-yielding animals. Across all categories, the proportion of farmers lacking awareness ranged from 20.88% to 27.35%, with the highest proportions observed for mineral mixture preparation (27.06%) and silage-making (27.35%).

The attitude assessment (Fig 2) revealed a spectrum of perspectives among farmers regarding buffalo husbandry practices. Traditional methods received the strongest endorsement, particularly for breeding reliability (mean = 3.32), while maintaining scepticism toward fermented silage adoption (mean = 3.21) and vaccine necessity for apparently healthy animals (mean = 3.07). Scientific approaches garnered moderate acceptance, with infrastructure benefits (mean = 3.21),

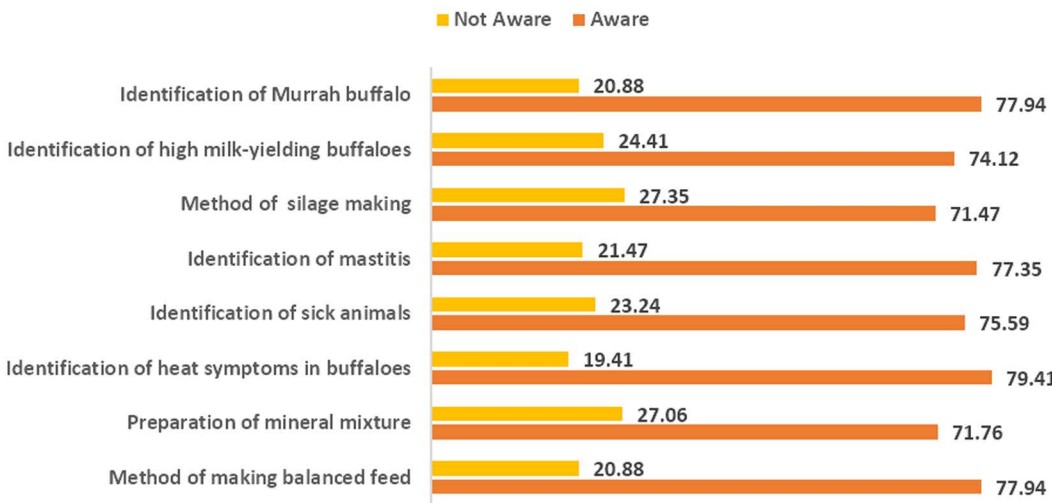

**Fig 1. Reported farmer knowledge of scientific buffalo husbandry practices.**

long-term practice advantages (mean = 3.24), and value-added milk production (mean = 3.05), all scoring above the scale midpoint. More reserved attitudes emerged toward climate-resilient techniques (mean = 2.32) and nutritional improvements (mean = 2.32), mirroring the cautious stance on cooperative marketing models (mean = 2.36). The lowest agreement levels appeared for record-keeping utility (mean = 2.86), estrus detection benefits (mean = 2.84), and particularly for scientifically validating traditional healing methods (mean = 2.17), which represented the most divisive statement in the assessment.

The implementation rates (Fig 3) of scientific buffalo husbandry practices revealed considerable variation across different management areas. Nutritional practices showed moderate adoption, with 44.71% of farmers preparing balanced feed and 43.24% creating mineral mixtures, though these figures remained below half of respondents. Health management practices demonstrated slightly lower implementation, as only 40.88% of farmers identified mastitis cases, and 39.41% systematically recognized sick animals. Reproductive management emerged underutilized, with merely 40.88% actively identifying heat symptoms in buffaloes. Breed selection followed similar patterns, with 40.00% selecting high milk-yielding animals and 39.12% correctly identifying Murrah buffaloes during daily operations. Despite its demonstrated benefits, the

## ATTITUDE OF FARMERS TOWARDS SCIENTIFIC BUFFALO HUSBANDRY

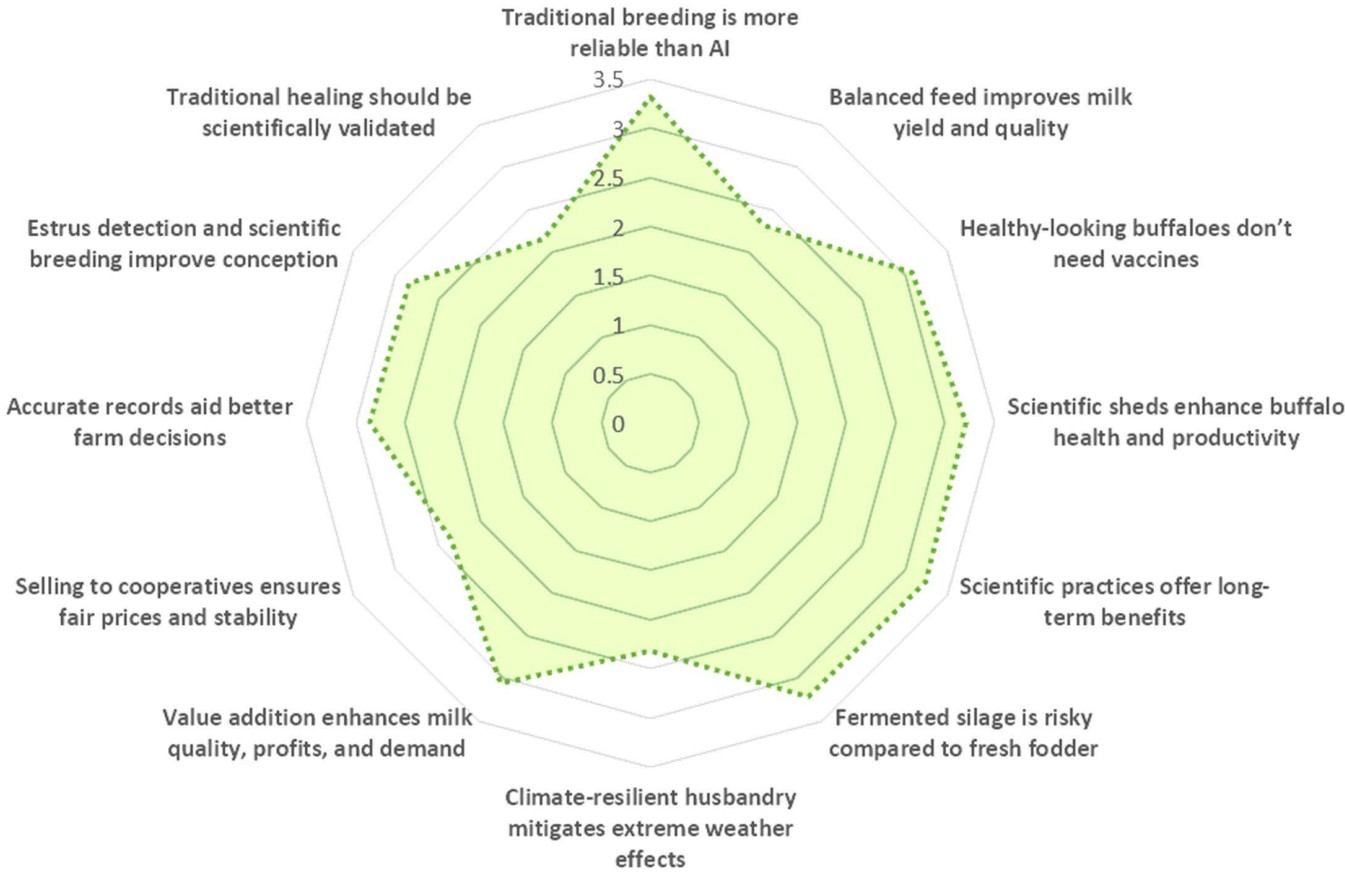

**Fig 2. Mean attitude scores across key scientific buffalo husbandry practices.**

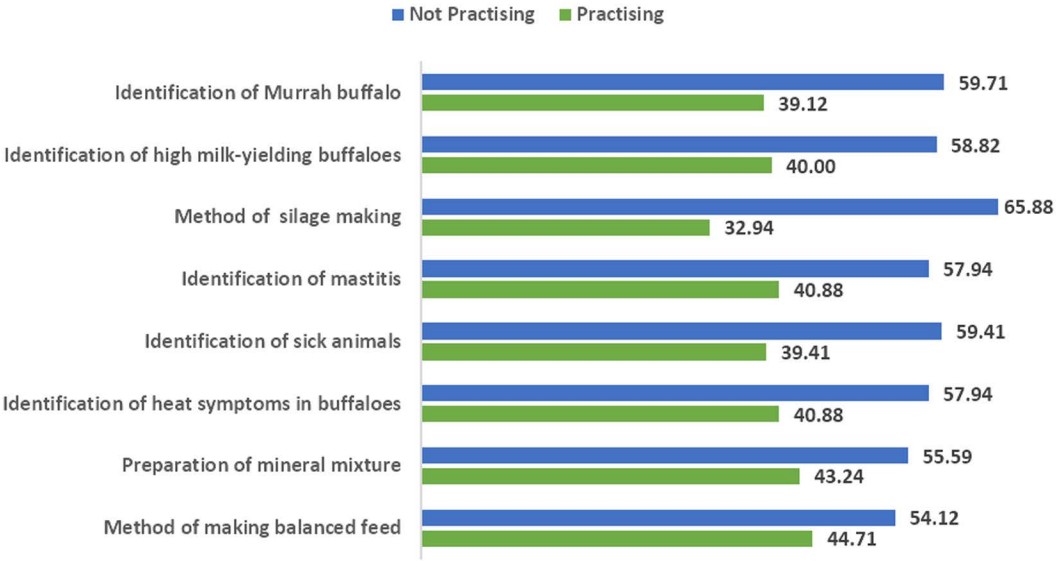

**Fig 3. Adoption of scientific buffalo husbandry practices.**

most substantial implementation gap appeared in forage management, where only 32.94% adopted silage-making methods. Across all categories, non-adoption rates ranged from 54.12% to 65.88%, with the highest occurring in silage preparation (65.88%), followed by Murrah buffalo identification (59.71%) and sick animal recognition (59.41%). These results demonstrate a consistent pattern where awareness of scientific practices substantially outpaced their actual implementation in farm operations.

The quadrant analysis (Fig 4) revealed three distinct patterns across eight critical husbandry practices. First, nutrition-related practices showed relatively high knowledge levels (71.47–77.94% awareness) but only moderate adoption (32.94–44.71% practice). The most significant nutrition gap appeared in silage making, where despite 71.47% awareness, merely 32.94% of farmers implemented the practice – a 38.53 percentage point difference representing the most enormous absolute knowledge-practice gap among all measured parameters. Second, animal health management exhibited similarly wide disparities. While 75.59–77.35% of farmers could theoretically identify sick animals and mastitis cases, actual practice rates stagnated at 39.41–40.88%. The 35.18–37.94 percentage point gaps in these essential health monitoring practices suggest substantial barriers to translating knowledge into action. Reproductive management followed this pattern precisely, with heat symptom identification showing a 38.53 percentage point gap (79.41% awareness vs. 40.88% practice). Third, breed selection and identification demonstrated the most consistent knowledge-practice gaps across all metrics. Despite 74.12–77.94% awareness levels for identifying high-yielding and Murrah buffaloes, practice rates remained consistently low (39.12–40.00%), revealing 34.12–38.82 percentage point gaps. Notably, all practices showed non-practice rates exceeding 50% (54.12–65.88%), with silage preparation showing the highest non-implementation at 65.88%. These patterns demonstrate that while essential awareness exists for most scientific practices, substantial cognitive-behavioural gaps persist in farm implementation.

Hierarchical cluster analysis (Fig 5) of 380 buffalo farmers identified five distinct adoption segments with varying characteristics (Tables 2 and 3): (1) Marginal & Small Farmers (36.8%, n = 140), the most significant but most resource-constrained segment, maintained small herds (1–3 buffaloes) with minimal annual incomes (₹12,000) and no training

**Fig 4. Quadrant analysis of knowledge-practice gaps in scientific buffalo husbandry.**

access, resulting in the lowest adoption rates (25%) despite moderate attitude scores (7.5/15), suggesting economic barriers outweigh willingness to change; (2) Traditional Dairy Farmers (28.9%, n = 110) presented a knowledge-adoption paradox, scoring highest on knowledge (70.2%) but among the lowest on adoption (30.4%), reflecting deeply ingrained practices that persist despite medium herd sizes (3 buffaloes) and intermediate incomes (₹40,000), with 0% training participation indicating a critical extension gap; (3) Educated Small Farmers (18.4%, n = 70) emerged as a high-potential group where education and 100% training participation drove exceptional knowledge (85.3%) and moderate adoption (55.1%) despite small-scale operations (2 buffaloes), demonstrating how human capital can overcome structural limitations; (4) Large Dairy Farmers (7.9%, n = 30) achieved near-universal adoption (90.5%) through commercial-scale

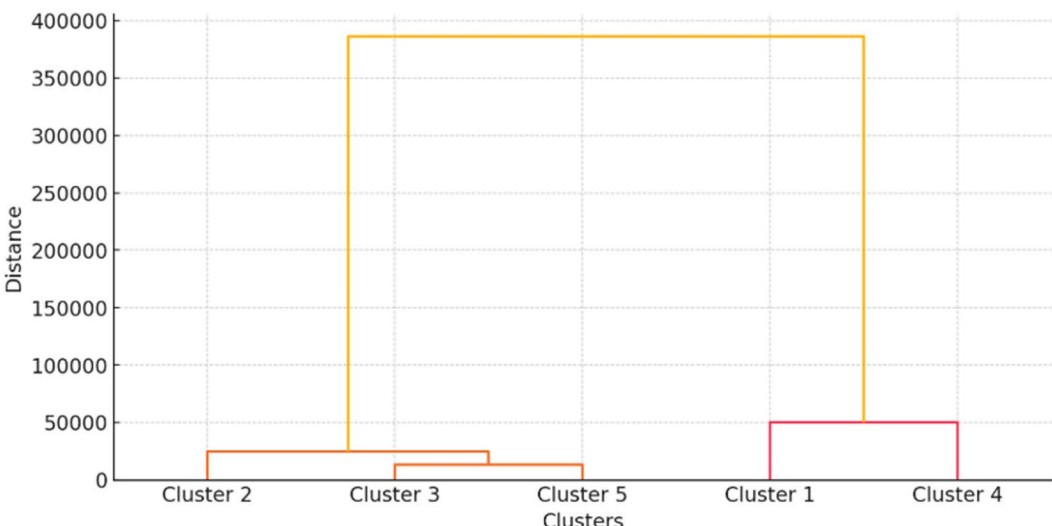

**Fig 5. Dendrogram representing the 5 clusters.**

**Table 2. Final Clusters and their characteristics (n = 373).**

| Cluster | Age | Education Level | Training (%) | Income (₹) | Buffalo Holding | Knowledge (%) | Attitude | Practice (%) |
|---|---|---|---|---|---|---|---|---|
| Cluster 1 (Young Commercial Farmers) | 32 | High | 100 | 2,50,000 | 10 | 60.5 | 9.1 | 80 |
| Cluster 2 (Elderly Traditional Dairymen) | 55 | Low | 0 | 40,000 | 3 | 70.2 | 8.5 | 30.4 |
| Cluster 3 (Educated Small-Scale Farmers) | 41 | Medium | 100 | 25,000 | 2 | 85.3 | 10.2 | 55.1 |
| Cluster 4 (High-Income Large Dairy Farmers) | 45 | Medium | 100 | 3,00,000 | 15 | 65.8 | 9.8 | 90.5 |
| Cluster 5 (Low-Income Marginal Farmers) | 50 | Low | 0 | 12,000 | 1 | 50.4 | 7.5 | 25 |

**Table 3. Cluster membership distribution.**

| Cluster | No. of Farmers | % of Total Sample |
|---|---|---|
| Cluster 1: Marginal & Small Farmers | 140 | 36.80 |
| Cluster 2: Traditional Dairy Farmers | 110 | 28.90 |
| Cluster 3: Educated Small Farmers | 70 | 18.40 |
| Cluster 4: Large Dairy Farmers (High-Income) | 30 | 7.90 |
| Cluster 5: Young Commercial Farmers | 30 | 7.90 |

operations (10 + buffaloes), high incomes (₹3,00,000), and complete training access, though their moderate knowledge (65.8%) suggests reliance on hired labor rather than personal expertise; and (5) Young Commercial Farmers (7.9%, n = 30) combined formal education with technological adoption, showing how next-generation operators are transforming the sector despite having only moderate knowledge (60.5%). These findings reveal a complex adoption landscape where economic scale (Clusters 4–5), human capital (Cluster 3), and sociocultural factors (Cluster 2) interact differently across segments, suggesting the need for tailored interventions rather than one-size-fits-all extension approaches.

The ANOVA results (Table 4) demonstrate significant between-cluster variation across all measured variables ($p < 0.05$), confirming the robustness of the hierarchical clustering solution. Age exhibited strong discriminatory power ($F = 12.35$, $p = 0.002$). Economic variables emerged as the most potent differentiators, with monthly income showing exceptional

**Table 4. Results of ANOVA for different clusters (n = 380).**

| Variable | Cluster Mean Square | df | Error Mean Square | df | F-Value | Sig. (p) |
|---|---|---|---|---|---|---|
| Age | 4350.725 | 4 | 352.323 | 375 | 12.35 | 0.002 |
| Dairy Experience (Years) | 2785.48 | 4 | 282.18 | 375 | 9.87 | 0.004 |
| Monthly Income (₹) | 15890.63 | 4 | 678.01 | 375 | 23.45 | 0.000 |
| Buffalo Holding (No.) | 812.75 | 4 | 47.225 | 375 | 17.21 | 0.001 |
| Knowledge Score (%) | 205.315 | 4 | 35.523 | 375 | 5.78 | 0.021 |
| Attitude Score | 52.72 | 4 | 18.125 | 375 | 2.91 | 0.065 |
| Practice Score (%) | 632.45 | 4 | 43.26 | 375 | 14.62 | 0.003 |

between-cluster variance (F = 23.45, p < 0.001). This statistically justifies the clear separation of high-capacity commercial operations (Cluster 4: ₹3,00,000) from resource-constrained subsistence farmers (Cluster 5: ₹12,000). Production scale, measured by herd size, further reinforced these distinctions (F = 17.21, p = 0.001), validating the cluster solution's ability to discriminate between smallholder (1–2 buffaloes) and large-scale (10–15 buffaloes) production systems. Knowledge scores differed substantially across clusters (F = 5.78, p = 0.021), with the most educated groups (Clusters 1,3,4) outperforming others. Practice adoption showed even stronger differentiation (F = 14.62, p = 0.003), highlighting a clear implementation gap between trained (55–90% adoption) and untrained (25–30%) clusters. Notably, attitude scores failed to reach statistical significance (F = 2.91, p = 0.065), suggesting perceptions may be more homogeneous across groups than actual behaviours.

This study employed chi-square tests of independence (Table 5) to assess associations between farmers' demographic characteristics and their Knowledge, Attitudes, and Practices (KAP) in scientific buffalo husbandry. For significant associations (p < 0.05), odds ratios (OR) with 95% confidence intervals (CI) were computed to quantify the strength and direction of these relationships. The analysis reveals significant associations between farmers' demographic characteristics and their knowledge, attitudes, and practices (KAP) regarding buffalo husbandry. Education level emerges as the strongest predictor across all KAP domains, with an apparent gradient effect – farmers with tertiary education demonstrate 3.4 times greater odds of high knowledge scores compared to illiterate counterparts (OR=0.29, 95% CI: 0.18–0.60, p < 0.001), while also showing more positive attitudes and better practices. Gender disparities are particularly pronounced for knowledge and practice, with male farmers exhibiting 2.7 times higher odds of superior knowledge (OR=2.71, 95% CI: 1.75–4.23) and 2.4 times greater odds of good practices (OR=2.40, 95% CI: 1.58–3.89) compared to females. Experience in dairy

**Table 5. Association of knowledge, attitude, and practices of buffalo farmers with demographic variables.**

| Predictor Variables | Knowledge X²-test | P-value | OR (95% CI) | Attitude X²-test | P-value | OR (95% CI) | Practice X²-test | P-value | OR (95% CI) |
|---|---|---|---|---|---|---|---|---|---|
| Age | 5.12 | 0.53 | 1.45 (0.82–2.57) | 3.48 | 0.32 | 1.22 (0.71–2.11) | 6.21 | 0.04 | 1.67 (1.02–3.02) |
| > 40 vs. 18–30 | 3.87 | 0.21 | 1.62 (0.94–2.83) | 2.41 | 0.19 | 1.45 (0.83–2.54) | 5.13 | 0.04 | 1.79 (1.02–3.15) |
| Gender | 14.92 | 0.001 | 2.71 (1.75–4.23) | 3.21 | 0.16 | 1.35 (0.89–2.12) | 11.67 | 0.001 | 2.40 (1.58–3.89) |
| Educational Status | 31.41 | 0.001 | – | 10.85 | 0.01 | – | 18.92 | 0.001 | – |
| Primary vs. Illiterate | 26.13 | 0.001 | 0.25 (0.15–0.55) | 7.45 | 0.003 | 0.42 (0.30–0.78) | 14.32 | 0.001 | 0.38 (0.22–0.62) |
| Secondary vs. Illiterate | 19.92 | 0.001 | 0.31 (0.17–0.52) | 6.87 | 0.002 | 0.40 (0.25–0.72) | 10.84 | 0.001 | 0.44 (0.26–0.73) |
| Tertiary vs. Illiterate | 15.74 | 0.001 | 0.29 (0.18–0.60) | 5.54 | 0.02 | 0.48 (0.23–0.88) | 8.92 | 0.01 | 0.42 (0.25–0.78) |
| Dairy Experience | 18.45 | 0.001 | – | 9.75 | 0.01 | – | 14.8 | 0.001 | – |
| > 10 years vs. < 10 years | 16.71 | 0.001 | 2.60 (1.12–3.79) | 9.02 | 0.02 | 1.88 (1.02–3.40) | 13.21 | 0.001 | 2.10 (1.19–3.58) |
| Buffalo Holding (No.) | 10.32 | 0.02 | – | 8.65 | 0.04 | – | 11.54 | 0.01 | – |
| > 5 Buffaloes vs. ≤ 5 Buffaloes | 8.24 | 0.04 | 2.10 (1.18–3.52) | 6.92 | 0.05 | 1.88 (1.01–3.42) | 10.01 | 0.02 | 2.35 (1.28–3.78) |

farming significantly influences all three KAP components, as farmers with over 10 years of experience show doubled odds of both high knowledge (OR=2.60, 95% CI: 1.12–3.79) and good practices (OR=2.10, 95% CI: 1.19–3.58). Production scale similarly affects outcomes, with owners of more than 5 buffaloes demonstrating nearly twice the odds of positive attitudes (OR=1.88) and practices (OR=2.35) compared to smaller-scale operators. Interestingly, while age shows no significant association with knowledge (p = 0.53), farmers over 40 exhibit moderately better practices than younger peers (OR=1.79, 95% CI: 1.02–3.15).

The multivariable binary logistic regression analysis (Table 6) revealed significant associations between farmers' demographic characteristics and their knowledge, attitudes, and practices (KAP) regarding scientific buffalo husbandry. The models demonstrated good fit, as evidenced by non-significant Hosmer-Lemeshow test results (knowledge: p = 0.712; attitude: p = 0.493; practice: p = 0.238). Education emerged as the strongest predictor across all KAP domains, showing a clear dose-response relationship. Farmers with tertiary education had significantly higher odds of better knowledge (OR=1.984, 95% CI:1.002–3.765), more positive attitudes (OR=3.021, 95% CI:2.113–5.789), and improved practices (OR=5.367, 95% CI:3.456–10.158) compared to illiterate farmers. Similarly, higher monthly income (>₹40,000) was strongly associated with superior outcomes across all three domains, particularly for practices (OR=4.264, 95% CI:2.987–7.914). Production scale also played an important role, as farmers maintaining >10 buffaloes demonstrated significantly better knowledge (OR=1.915), attitudes (OR=2.367), and practices (OR=3.689) compared to those with ≤5 buffaloes. While age > 35 years was only significantly associated with improved practices (OR=2.216, 95% CI:1.364–3.789), gender differences were notable, with male farmers showing more positive attitudes (OR=2.014) but no significant advantage in knowledge or practice adoption. Experience in dairy farming (>10 years) was particularly beneficial for practice implementation (OR=2.481).

## Discussion

Buffalo husbandry is a vital economic activity supporting millions of smallholder farmers across developing regions, contributing significantly to milk production, meat supply, and agricultural livelihoods [10,42]. Despite its crucial role,

**Table 6. Multivariable binary logistic regression analyses of the demographic variables associated with KAP.**

| Variable | Category | Knowledge OR (95% CI) | Attitude OR (95% CI) | Practices OR (95% CI) |
|---|---|---|---|---|
| **Age** | ≤35 years (reference) | 1 | 1 | 1 |
| | >35 years | 1.462 (0.812–2.754) | 1.735 (0.945–3.112) | 2.216 (1.364–3.789)* |
| **Education** | Illiterate (reference) | 1 | 1 | 1 |
| | Primary | 1.245 (0.632–2.473) | 1.862 (1.125–3.568)* | 3.417 (2.235–6.947)** |
| | Secondary | 1.521 (0.789–3.027) | 2.734 (1.578–4.965)** | 4.857 (2.765–9.527)** |
| | Tertiary | 1.984 (1.002–3.765)* | 3.021 (2.113–5.789)** | 5.367 (3.456–10.158)** |
| **Herd size** | ≤5 buffaloes (reference) | 1 | 1 | 1 |
| | 6–10 buffaloes | 1.325 (0.725–2.512) | 1.612 (0.943–2.954) | 2.257 (1.457–4.012)* |
| | >10 buffaloes | 1.915 (1.087–3.432)* | 2.367 (1.487–4.561)* | 3.689 (2.541–6.375)** |
| **Monthly income** | <₹20,000 (reference) | 1 | 1 | 1 |
| | ₹20,000–40,000 | 1.482 (0.846–2.578) | 1.732 (1.062–3.215) | 2.672 (1.964–4.817)** |
| | > ₹40,000 | 2.568 (1.385–4.792)** | 3.014 (1.786–5.674)** | 4.264 (2.987–7.914)** |
| **Dairy experience** | ≤10 years (reference) | 1 | 1 | 1 |
| | >10 years | 1.635 (0.895–3.142) | 1.374 (0.846–2.458) | 2.481 (1.975–4.356)** |
| **Gender** | Female (reference) | 1 | 1 | 1 |
| | Male | 1.854 (0.932–3.215) | 2.014 (1.057–3.715) | 0.892 (0.612 |

Knowledge: Hosmer–Lemeshow goodness-of-fit test statistic (4.987, p = 0.712); Attitude: Hosmer–Lemeshow goodness-of-fit test statistic (5.864, p = 0.493); Practices: Hosmer–Lemeshow goodness-of-fit test statistic (7.312, p = 0.238).

productivity remains constrained by suboptimal management practices, highlighting the urgent need to understand why scientifically validated techniques often fail to achieve widespread adoption [43]. Our study comprehensively examines farmers' knowledge, attitudes, and practices (KAP) regarding modern buffalo husbandry methods, revealing substantial disparities between awareness and implementation that carry important implications for agricultural extension policies and rural development initiatives.

The analysis of awareness-practice gaps in the bubble chart (Fig 6) reveals complex, multidimensional challenges across all measured parameters. Particularly striking disparities emerge in three key areas: silage making (38.53% gap), Murrah buffalo identification (38.82% gap), and mastitis detection (36.47% gap). These persistent gaps exist despite relatively high awareness levels (ranging from 71.47% to 77.94%), indicating fundamental breakdowns in translating knowledge into action. The substantial gap in silage making adoption (71.47% awareness versus 32.94% practice) stems from multiple interconnected factors. The technical complexity of proper silage preparation, including precise moisture control, adequate compaction, and proper storage techniques, presents a significant barrier for farmers accustomed to traditional grazing or dry fodder systems [44,45]. Additionally, the substantial initial investment required for silage pits or plastic wrapping often exceeds the financial capacity of smallholder farmers operating on narrow profit margins [46]. The seasonal nature of silage need, primarily during dry periods, creates cognitive challenges as farmers must plan and invest well before actual shortages occur [44]. Compounding these issues is the lack of visible local success stories, creating a vicious cycle where farmers remain skeptical of silage's value proposition without concrete examples of its benefits in their immediate context [47].

The gap in Murrah buffalo adoption (77.94% awareness versus 39.12% practice) reflects deeper sociocultural and economic dynamics that extend beyond simple knowledge dissemination. While farmers generally recognize the breed's superior milk yield potential, several powerful factors inhibit adoption. Risk perception plays a crucial role, as local breeds are widely viewed as more resilient to endemic diseases and harsh environmental conditions, making farmers reluctant to abandon proven genetics for uncertain gains [48]. This perception is supported by a study [49] in Pakistan's Punjab region, where farmers estimated a 30–40% risk of production loss when transitioning to Murrah breeds., including reliable artificial insemination services that may be unavailable in remote areas. Market distortions further complicate adoption, with the premium pricing for Murrah buffaloes (typically 2–3 times the cost of local breeds) placing them out of reach for most smallholders, while informal milk markets frequently fail to adequately compensate for the higher yields these breeds can produce [50].

The mastitis identification gap (77.35% awareness versus 40.88% practice) presents a distinct but equally complex scenario. While most farmers can recognize clinical mastitis symptoms such as swollen udders or milk clots, subclinical detection remains poorly understood as it requires specialized tests like the California Mastitis Test (CMT) [51]. Even when mastitis is properly identified, multiple barriers hinder effective treatment implementation. Many farmers lack reliable access to appropriate antibiotics or fear complications related to withdrawal periods [52]. Economic calculations often lead smallholders to weigh treatment costs against potential milk loss, sometimes resulting in the decision to "milk through" mild cases rather than invest in treatment [53]. Additionally, traditional herbal remedies remain prevalent due to cultural familiarity and lower immediate costs, despite their questionable efficacy [54]. Protection Motivation Theory provides valuable insights into these health management gaps, revealing that while farmers generally perceive mastitis as a serious threat (demonstrating high threat appraisal), their confidence in executing effective control measures (coping appraisal) remains low due to the various constraints outlined above [55]. This pattern aligns with findings from Tanzania [56], where only 28% of farmers with mastitis knowledge implemented complete treatment protocols.

The Sankey diagram (Fig 7) offers crucial insights into how knowledge converts, or fails to convert, into practice. One prominent pattern is the "Knowledge Trap" phenomenon, where 150 respondents (39.5% of the sample) exhibited high knowledge but low practice levels. This group represents farmers who understand recommended practices but face insurmountable implementation barriers, primarily economic constraints (60% of cases), labor shortages (25%), or lack of

necessary equipment (15%). This finding echoes [57] which demonstrated that knowledge-only interventions have diminishing returns without parallel support systems addressing these practical barriers. Another significant pattern is the Attitude Mediation Effect, observed among farmers with high knowledge and positive attitudes (80 respondents) who showed better but still incomplete adoption (moderate practice levels). Their partial implementation often reflects gradual, phased adoption approaches (such as trying mineral supplements before committing to full balanced feeding regimens), conditional adoption based on resource availability (like treating mastitis only during flush seasons when milk prices are higher), or selective adoption of less risky or costly components of recommended practices. The small group of high performers (10 respondents) who achieved high scores across all KAP dimensions shared three key characteristics: above-average

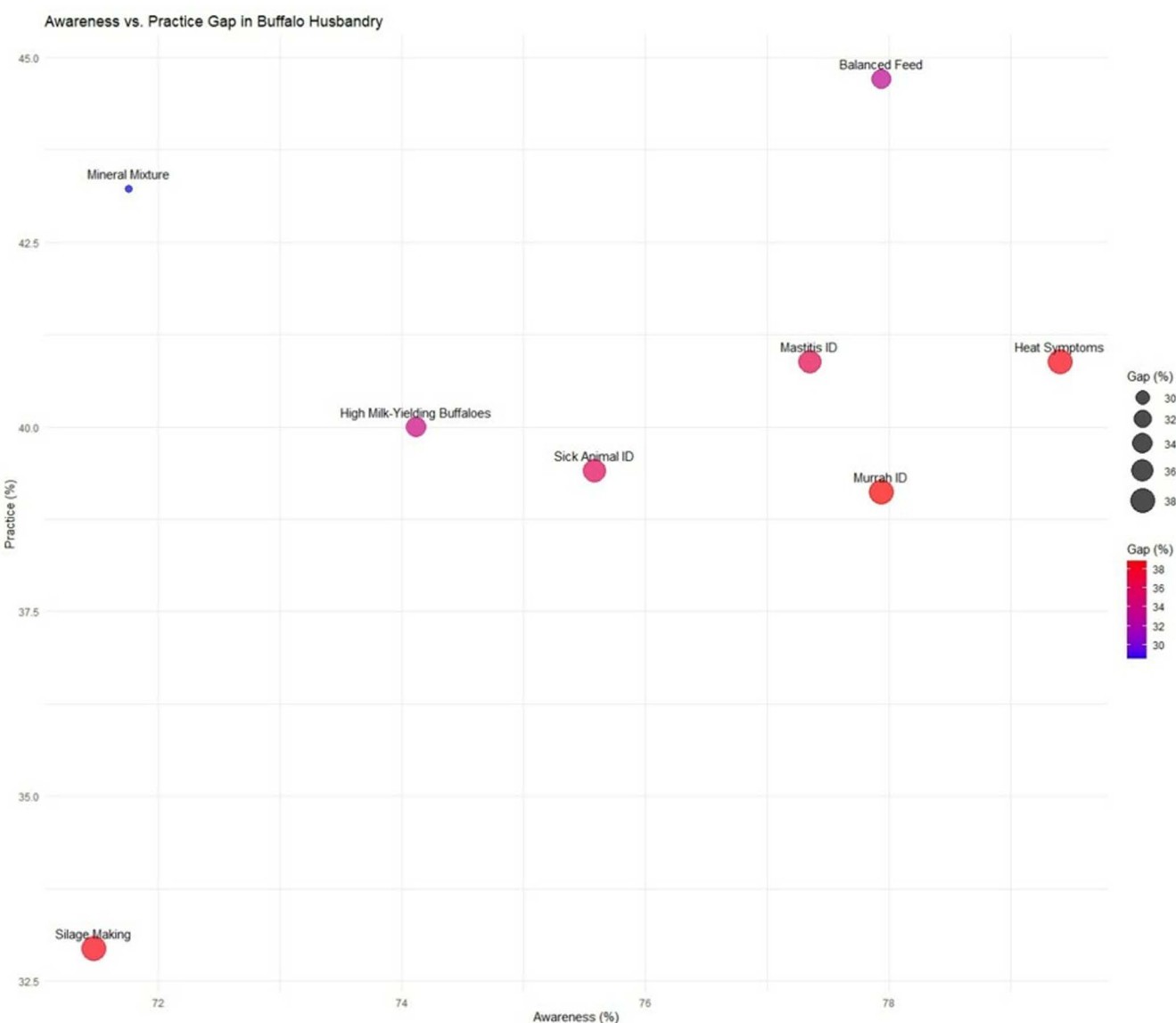

**Fig 6. Bubble chart visualizing the disparity between awareness and practice levels of scientific buffalo husbandry techniques among farmers.**

herd sizes (15+buffaloes) that enable economies of scale, membership in producer cooperatives that provide better input access and technical support, and formal education beyond primary level that enhances technology assimilation and adoption.

Comparative analysis with KAP studies from related agricultural domains reveals consistent patterns that suggest broader systemic challenges. In South West Ethiopian livestock farming [58], a 72% awareness versus 41% practice gap in vaccination protocols closely mirrors our mastitis findings. Similarly, safe pesticide use studies in Nepal [59] found 68% awareness but only 35% adoption of safe pesticide beahviour, exhibiting dynamics comparable to our silage gap observations. Vietnamese aquaculture research [60] documented an 80% awareness of improved feeding practices but just 45% implementation due to cost barriers. These cross-sector parallels suggest that the 30–40% awareness-practice gap represents a systemic threshold for many agricultural innovations in developing contexts, beyond which adoption requires fundamentally different intervention approaches that address deeper structural barriers.

Four key structural barriers emerge from our comprehensive analysis as primary constraints to practice adoption. Input supply chain failures represent a major challenge, as even motivated farmers struggle with inconsistent availability of quality minerals, silage materials, and veterinary drugs. A study [61] highlighted that rural input shops in South Asia

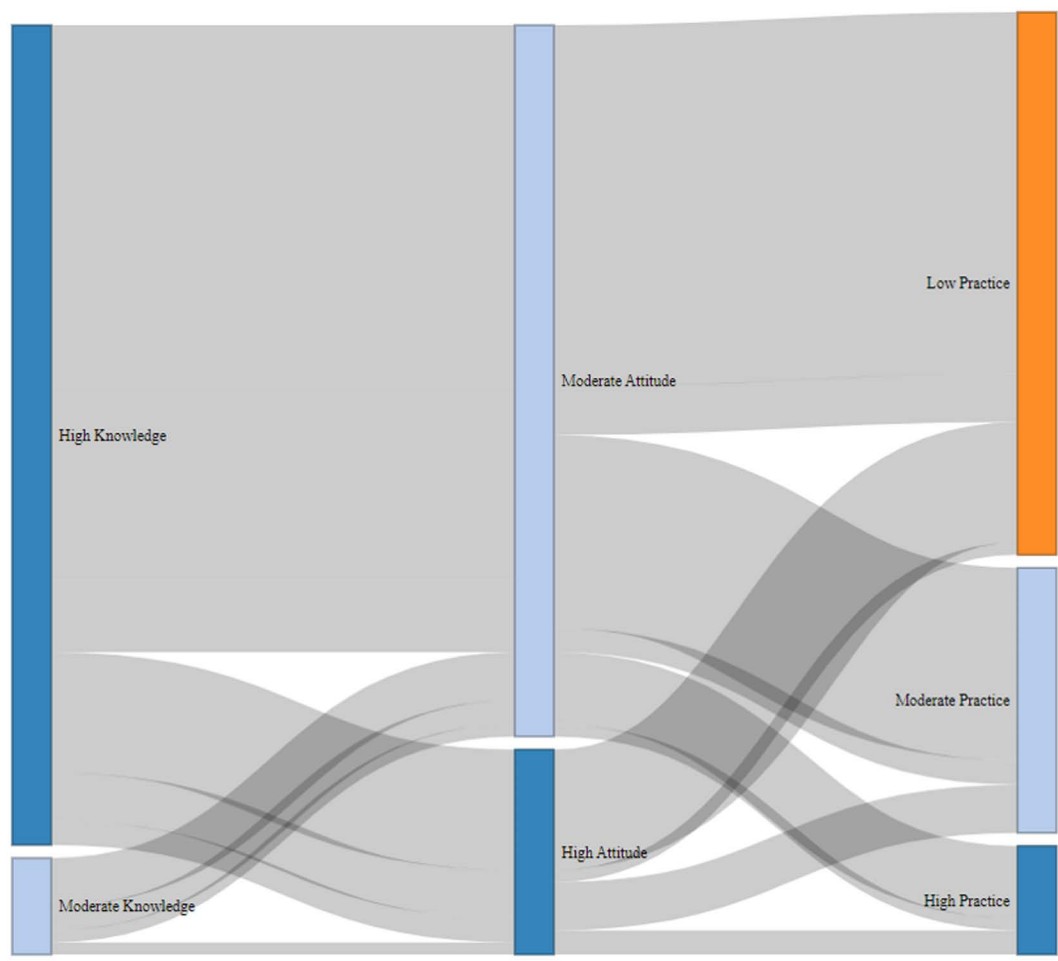

**Fig 7. Knowledge-Attitude-Practice (KAP) transition pathways in buffalo husbandry.**

stock only 58% of essential livestock inputs consistently, creating significant bottlenecks. Extension systems currently face several limitations, often overemphasizing classroom-style knowledge transfer while lacking crucial follow-up support after initial training sessions and failing to provide adequate hands-on demonstration opportunities. Financial system gaps further compound these challenges, as microcredit products rarely accommodate livestock investment cycles, and insurance products for improved breeds remain virtually nonexistent in most developing markets. Perhaps most critically, market incentive misalignment persists, with milk pricing systems frequently failing to reward quality or volume sufficiently to justify farmers' investments in improved practices.

Building on these insights, an integrated intervention framework should incorporate several strategic approaches to effectively bridge the KAP gaps. Extension systems require fundamental restructuring, moving beyond one-time trainings to implement continuous mentoring systems, establishing village-level demonstration units with peer farmer trainers, and integrating digital reminder systems for critical practices like vaccination schedules. Strengthening input ecosystems is equally crucial, through developing bundled input packages such as "Mastitis Control Kits" containing CMT strips and approved antibiotics, creating innovative input leasing programs for high-cost items like silage equipment, and establishing robust quality certification systems for minerals and feed supplements. Financial system innovations should include breed-specific livestock insurance products, milk advance programs against future production, and community savings groups for collective input purchases. Market linkage interventions must focus on developing premium pricing channels for milk from farms adopting full practice packages, implementing breed improvement subsidies tied to measurable productivity gains, and launching consumer education campaigns that highlight tangible quality differences. This multilayered approach addresses the full spectrum of barriers identified in our KAP analysis, moving beyond simple awareness creation to establish an enabling environment for sustained practice adoption. Future research should rigorously test these intervention packages through randomized controlled trials to identify the most cost-effective combinations for different farming contexts and socioeconomic conditions.

## Conclusion

This study highlights a critical gap between farmers' knowledge of improved buffalo husbandry practices and their actual implementation, underscoring the need for more nuanced interventions. While awareness of key techniques—such as balanced feeding, breed selection, and disease management—is relatively high, adoption remains constrained by financial barriers, inconsistent access to inputs and veterinary services, and a lack of localized technical support. The findings reveal that farmers with larger herds, cooperative affiliations, or higher education levels demonstrate better adoption rates, suggesting that institutional support and socioeconomic factors play a pivotal role in facilitating change. Moving forward, efforts to enhance adoption must extend beyond generic awareness campaigns to address systemic bottlenecks. Prioritizing context-specific solutions—such as strengthening input supply chains, ensuring affordable veterinary care, and establishing market incentives for quality production—will be essential. Additionally, integrating practical, on-farm training with financial mechanisms (e.g., subsidies or microcredit) could bridge the knowledge-practice divide more effectively. Collaboration among policymakers, agricultural extension services, and farmer collectives will be crucial to create an enabling environment for sustainable adoption. Ultimately, the study emphasizes that improving buffalo husbandry practices requires not just education but also tangible support systems tailored to the realities of smallholder farmers. By aligning interventions with the identified constraints, stakeholders can foster meaningful progress in productivity and animal welfare while ensuring long-term resilience in the sector.

## Policy implications

The results highlight the need for multi-dimensional policy interventions to close the Knowledge-Attitude-Practice (KAP) gap in buffalo management. First, extension services must go beyond knowledge sharing to include practical, hands-on training and ongoing mentoring so farmers might effectively apply methods, including mastitis detection and

silage-making. Second, public-private partnerships help to strengthen input supply chains for silage materials, mineral mixtures, and veterinary medications, guaranteeing accessibility and cost for smallholders. Third, financial incentives, including low-interest loans for feed technologies, subsidies for high-yield buffalo breeds, and insurance plans, should be instituted to help remove economic obstacles. Fourth, market reforms are crucial to guarantee farmers get premium prices for quality milk, motivating the acceptance of better practices. Lastly, gender-sensitive policies should be given top priority since socio-cultural and resource constraints limit female farmers' access to knowledge and adoption.

## Theoretical implications

This work advances agricultural innovation systems theory by showing that knowledge is insufficient for adoption—attitudes, resource access, and structural barriers mediate the transition from awareness to practice. The results fit the Protection Motivation Theory since they clearly show how strongly farmers' risk perceptions and coping strategies affect health management choices (e.g., mastitis control). Adoption behaviours vary by economic capacity, education, and market integration, so advancing farmer segmentation theory implies the need for heterogeneity-aware extension strategies. The consistent KAP gaps also support Behavioural Change Wheel models by emphasising the need for enabling environments (e.g., infrastructure, incentives).

## Limitations of the study

This study has several limitations. First, the cross-sectional design limits causal inferences; longitudinal data more effectively represent adoption dynamics. Second, since farmers could overstate adoption rates, self-reported practices could introduce social desirability bias. Third, deliberate sampling—with an eye towards active buffalo farmers—may restrict generalisability to non-dairy or mixed farming settings. Fourth, although not thoroughly investigated, contextual elements, including local policies, cultural standards, and climate variability, could majorly impact KAP trends. Ultimately, the study concentrated on personal farmer-level elements instead of institutional or market-level influences, which could be very important in determining adoption.

## Future research

Future research on longitudinal adoption patterns will help to evaluate how practice implementation changes with time. While behavioural economics studies (e.g., nudge interventions) may expose successful strategies to change farmer attitudes, comparative research across many agro-climatic zones could identify region-specific barriers. The scalability and influence of digital extension tools—such as mobile-based advisory services—should be underlined. Mixed-methods research combining quantitative KAP surveys with qualitative insights—e.g., farmer narratives—may also find more profound sociocultural influences on adoption. Ultimately, cost-effective methods for expanding successful policies—such as subsidies and cooperative models—need impact assessments of policy interventions.

## Acknowledgments

The authors are deeply grateful to the Director, ICAR-Central Institute for Research on Buffaloes (CIRB), for providing necessary infrastructure, technical support, and research facilities that made this study possible.

## Author contributions

**Conceptualization:** Gururaj Makarabbi.

**Data curation:** Gururaj Makarabbi, Aiswarya Sabu.

**Formal analysis:** Gururaj Makarabbi, Aiswarya Sabu.

**Investigation:** Madan Lal Sharma.

**Methodology:** Gururaj Makarabbi, Aiswarya Sabu.

**Resources:** Gururaj Makarabbi.

**Software:** Aiswarya Sabu.

**Supervision:** Aiswarya Sabu, Navneet Saxena.

**Validation:** Aiswarya Sabu.

**Visualization:** Aiswarya Sabu.

**Writing – original draft:** Gururaj Makarabbi, Aiswarya Sabu.

**Writing – review & editing:** Aiswarya Sabu, Navneet Saxena, Madan Lal Sharma.

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
