## [Decision Letter · Decision Letter 0]

An Analysis of Knowledge, Attitude and Practice Gaps in Scientific Buffalo Husbandry

PONE-D-25-18442

Dear Dr. Aiswarya,

We’re pleased to inform you that your manuscript has been judged scientifically suitable for publication and will be formally accepted for publication once it meets all outstanding technical requirements.

Kind regards,

Julio Cesar de Souza, Ph.D.

Academic Editor

PLOS ONE

Additional Editor Comments (optional):

Considering that both reviewers accepted the paper for publication, I am in favor of it!

Publish it!!

Reviewers' comments:

Reviewer's Responses to Questions

**Comments to the Author**

1. Is the manuscript technically sound, and do the data support the conclusions?

Reviewer #1: Yes

Reviewer #2: Yes

2. Has the statistical analysis been performed appropriately and rigorously?

Reviewer #1: Yes

Reviewer #2: Yes

3. Have the authors made all data underlying the findings in their manuscript fully available?

Reviewer #1: Yes

Reviewer #2: Yes

4. Is the manuscript presented in an intelligible fashion and written in standard English?

Reviewer #1: Yes

Reviewer #2: Yes

Reviewer #1: Improve the linguistic errors

title: An Analysis of Knowledge, Attitude and Practice Gaps in Scientific Buffalo Husbandry

PONE-D-25-18442

Despite its crucial role in India's dairy industry, buffalo husbandry faces productivity challenges due to low adoption rates of key practices. A study of 380 Indian buffalo farmers revealed significant gaps between their knowledge, attitudes, and practices (KAP). While awareness of heat detection (79.4%), mastitis identification (77.4%), and balanced feeding (77.9%) was substantial, actual implementation ranged from only 32.9% to 44.7%, with silage-making showing the largest discrepancy (71.5% awareness vs. 32.9% adoption). Cluster analysis identified five distinct farmer groups with varying adoption levels, from smallholders (36.8% of farmers) practicing only 25% of techniques despite reasonable knowledge to commercial farmers (7.9%) achieving 90.5% adoption.

Reviewer #2: The manuscript seems to be tecnically sound, the statistical analysis have been performed appropriately,the authors made all data full available, the language is clear, The conclusions fully reflect what is reported in the text of the work

**Do you want your identity to be public for this peer review?** For information about this choice, including consent withdrawal, please see our Privacy Policy

Reviewer #1: **Yes: ** Sameh A Abdelnour

Reviewer #2: No
